# Inflammation Causes Exacerbation of COVID-19: How about Skin Inflammation?

**DOI:** 10.3390/ijms232012260

**Published:** 2022-10-14

**Authors:** Mayumi Komine, Tuba Mussarat Ansary, Md Razib Hossain, Koji Kamiya, Mamitaro Ohtsuki

**Affiliations:** Department of Dermatology, Jichi Medical University, 3311-1 Yakushiji, Shimotsuke 329-0498, Tochigi, Japan

**Keywords:** atopic dermatitis, COVID-19, inflammation, psoriasis, skin, SARS-CoV-2

## Abstract

COVID-19 is a recently emerged viral infection worldwide. SARS-CoV-2, the causative virus, is believed to have emerged from bat coronaviruses, probably through host conversion. The bat coronavirus which has the highest gene homology to SARS-CoV-2 specifically infects deep forest bats in China whose habitat extends through the Middle East to Southern Europe. Host conversion might have occurred due to the deforestation by humans exposing wild bats to the environment they had never encountered before. SARS-CoV-2 infects cells through two mechanisms: through its receptor ACE2 with the help of enzyme TMPRSS and through membrane fusion with the help of elastases in the inflammatory condition. Obesity, hypertension, diabetes mellitus, and pulmonary diseases cause poor prognosis of COVID-19. Aging is another factor promoting poor prognosis. These diseases and aging cause low-level and persistent inflammation in humans, which can promote poor prognosis of COVID-19. Psoriasis and atopic dermatitis are the major inflammatory skin diseases. These inflammatory skin conditions, however, do not seem to cause poor prognosis for COVID-19 based on the epidemiological data accumulated so far. These mechanisms need to be elucidated.

## 1. Introduction

COVID-19 (coronavirus disease 2019) is a recently emerged viral infection caused by severe acute respiratory syndrome coronavirus 2 (SARS-CoV-2). More than 450 million people have been infected worldwide, with more than 6 million deaths and almost 444 million people surviving but many continuing to suffer from subsequent symptoms [1]. SARS-CoV-2 emerged from bat coronaviruses in the course of host conversion by acquiring the ability to infect humans. The emergence of novel viruses is thought to increase in the future because of human activity invading wild animal habitats.

It has been known in the early stage of infection that inflammation exacerbates its severity. Patients characterized by old age, diabetes mellitus, respiratory diseases, metabolic syndrome, or obesity have bad prognosis compared to those without these complications. Many aspects of SARS-CoV-2 are similar to those of the SARS coronavirus, such as receptors and host cell entrance mechanisms. On the other hand, patients with psoriasis or atopic dermatitis, the major inflammatory skin diseases, seem to have less severe infection compared to the population without these skin diseases.

This review covers the basic nature of SARS-CoV-2, including the possible emergence pathways, mechanisms of infection and exacerbation in patients with inflammation, and the discussion why psoriasis and atopic dermatitis patients tend to have less severe infection compared to the population without skin diseases.

## 2. SARS-CoV-2

SARS-CoV-2 is a novel coronavirus that emerged from bat coronaviruses which acquired an ability to infect humans in 2019. It has been believed that the first patient appeared in December 2019 in Wuhan City, Hubei Province, in Central China. However, some recent reports demonstrate evidence implying its existence earlier than widely believed [2]. SARS-CoV-2 is the seventh member of the coronavirus family to infect humans. The common cold is usually caused by four varieties of coronaviruses (HCoV-229E, HCoV-NL63, HCoV-OC43, and HCoV-HKU1) which evoke a wide range of upper respiratory infection symptoms. MARS-COV and SARS-CoV are responsible for severe atypical pneumonia and utilize dipeptidyl peptidase 4 (DPP4) and angiotensin-converting enzyme 2 (ACE2), respectively, as their receptors to enter human cells distributed abundantly in the lower respiratory tract in humans [3]. The SARS-CoV-2 genome sequence shares 70% similarity with SARS-CoV and utilizes the same receptor as SARS-CoV, but with a 10-to-20-fold higher affinity, which makes human-to-human transmission easier [4].

The origin of SARS-CoV-2 is still uncertain, but it is believed that it originated from bat coronaviruses. Bats with thousands of gene types live deep in the mountains and forests of China, and various subspecies of coronaviruses infect specific genetic types of bats. The specificity has been strictly conserved; however, the specificity loosens when the environment changes, such as in case of the virus–bat relationship, resulting in host conversion [5,6,7]. The genetic state is very unstable when host conversion occurs, which might have allowed coronaviruses to infect humans when otherwise they had not. Among them, RatG13 is the coronavirus subspecies with the highest genetic homology with SARS-CoV-2 [2]. This virus specifically infects greater horseshoe bats (*Rhinolophus ferrumequinum*) which live in wide areas in China and are dispersed throughout the Middle East to Southern Europe. The coronavirus detected in this species of bats in the deep forests of China is genetically very similar to SARS-CoV-2 with an overall 96.2% genome sequence identity. The S gene, which encodes the receptor binding the spike protein of the coronavirus, is highly divergent between various coronaviruses. The other SARS-related coronaviruses showed a less than 75% sequence identity, while RatG13 showed an almost 93% identity with the SARS-CoV-2 S gene sequence [2].

There might have been an intermediate host which mediated the SARS-CoV-2 infection from bat to humans. In SARS-CoV, the palm civet is one possible intermediate host, and for MARS, the dromedary is thought to be the mediator. In SARS-CoV-2, the pangolin is thought to be the possible intermediate host [6]. Pangolins are the most poached mammals in the world according to the World Wide Fund for Nature (WWF) (Pangolin|Species|WWF, worldwildlife.org, accessed on 9 September 2022). Their skins are traded at high prices, and many poachers go deep into undeveloped forests to capture these animals with hunting dogs. This is another way of changing the environment of native habitat of pangolins and bats, exposing them to animals and other creatures they have never encountered before. 

SARS and SARS-CoV-2 share two pathways to infect cells; one is through ACE2 to the endosome pathway utilizing TMPRESS2 and the other is through cell membrane fusion in inflammatory condition utilizing elastases produced by environmental neutrophils.

## 3. Inflammation Exacerbates COVID-19 Infection

SARS-CoV-2 infection in most patients takes a mild course similarly to common cold, but in certain populations of patients, it takes a radical course, causing severe pneumonia, and sometimes requires ventilators or extracorporeal membrane oxygenation (ECMO). It is believed that this virus in certain population evokes a cytokine storm which cannot be controlled by regular anti-inflammatory medications. This biphasic aspect of COVID-19 has been known since the early stages of the pandemic: the first phase as the viral infection phase and the latter phase as the inflammation (cytokine storm) phase [8,9] (Figure 1). The treatments targeting these two phases should differ, with antiviral agents such as remdesivir, drugs targeting viral infection, and anti-inflammatory agents such as tocilizumab (anti-IL-6 receptor antibody) and glucocorticoids targeting cytokine storm. 

The clinical course of COVID-19 is divided into three stages. Stage 1 is a viral infection phase with mild common-cold like symptoms. In stage 2, most patients with COVID-19 recover, but a small portion of patients have sustained or even enhanced inflammatory condition and progress into stage 3. In stage 3, various inflammatory cytokines are systemically produced and patients suffer from severe respiratory distress syndrome with cytokine storm. Some patients show coagulopathy.

Many epidemiological surveys revealed that patients with diabetes mellitus, high blood pressure, respiratory disease, or obesity have a poorer prognosis compared to those without these conditions [10,11,12]. COVID-19 patients with these comorbidities are at a higher risk of developing severer pneumonia with cytokine storm and requiring artificial respirators or ECMO.

The reason why the disease exacerbates or becomes severe in certain populations has not been fully elucidated. However, inflammation seems to cause a bad prognosis because these comorbidities share common pathophysiology of low level but a persistent chronic inflammatory state.

SARS-CoV-2 enters host cells by biding to its receptor angiotensin converting enzyme ACE2 with spike protein (S). Transmembrane protease, serine (TMPRSS) 2, is one of the proteases which can prime the SARV-CoV and SARS-CoV-2 spike protein capable of binding their receptor ACE2 (Figure 2). The inhibition of this enzyme is one way to inhibit SARS-CoV-2 infection to cells [13]. Upon entering cells, ACE2 is internalized, resulting in reduction in ACE2 expression. ACE2 works competitively with ACE; ACE enhances inflammation as well as increases blood pressure and myocardial contraction while ACE2 has anti-inflammatory properties, suppressing inflammation as well as decreasing blood pressure and myocardial contraction. The ACE2 gene localizes on X chromosomes with a higher expression in females, and a lower severity of ARDS caused by the viral infection in female patients has been known. Massive COVID-19 infection could exacerbate inflammation by reducing the expression level of ACE2 by internalization of this receptor [10,14].

SARS-CoV-2 and SARS-CoV take another route to infect cells when there are abundant elastases in the surrounding environment: the cell membrane of SARS-CoV-2 fuses with that of the host cell and the viral genome is released into the host cells directly establishing the infection. This type of infection allows SARS-CoV-2 to grow and proliferate more rapidly, which could make its infection severer. Thus, an inflammatory environment, such as chronic occlusive pulmonary disease (COPD) lung with neutrophil infiltration, could be more vulnerable to severe COVID-19 pneumonitis than healthy conditions (Figure 3) [13].

Patients with high blood pressure are always under a higher mechanical stress because their heart and blood vessels endure a higher mechanical force. This kind of situation renders patients’ cardiovascular systems vulnerable to tissue damage by inflammation provoked by mechanical force. The low-level but chronic inflammation in the cardiovascular system causes infiltration of inflammatory macrophages and neutrophils in tissues which may make them susceptible to severe COVID-19 [9]. Recent findings reveled that the renin–angiotensin systems (RAS) pathway, the main endocrine pathway causing hypertension, is involved in COVID-19 inflammation [15]. The internalization of the virus downregulates the ACE2 expression on the cell surface, which causes elevation of angiotensin II and subsequent overactivation of cardiovascular systems, inducing vasoconstriction, increase in blood pressure, and thrombosis. Recent studies revealed that hypertensive heart injury is dependent on the dysregulation of autophagy, which is considered a novel therapeutic target of cardiovascular diseases. Angiotensin II has been shown to induce an increase in autophagy, and in vivo treatment of an autophagy inhibitor decreased blood pressure and improved endothelium-dependent relaxation. MAPK8 (JNK1) is involved in autophagy by phosphorylation of BCL2 (B cell leukemia/lymphoma 2), disrupting the interaction of BCL2 and BECN1 (beclin 1), resulting in BECN1-dependent autophagy [16]. A recent network pharmacology approach revealed the novel therapeutic targets of COVID-19 inflammation, including MAPK8 (JNK1), MAPK10 (JNK3), and BAD (Bcl-2-associated death promoter), which are involved in the RAS signaling pathway and autophagy. The inhibitors of these molecules, by decreasing RAS activation and autophagy, downregulate the cardiovascular burden, vasoconstriction, and thrombosis [15].

### 3.1. Aging-Related Poorer Prognostic Outcomes in COVID-19

The most important and enigmatic condition in the COVID-19 infection is the steeply increased mortality rate in the elderly. Elderly patients, who are the most susceptible population to severe COVID-19 infection, are often associated with multiple comorbidities such as cardiopulmonary diseases, diabetes mellitus, hypertension, and obesity. Aging itself is the process of chronic inflammation with increased senescence cells causing enhanced inflammation when lysed by microbes. COVID-19 often causes sepsis in this population with profound and persistent hypoxia, hypercoagulability, and altered aerobic glycolytic metabolism. Lungs, liver, kidneys, heart, brain, and gastrointestinal tract are frequently affected, leading to hypoxia caused by respiratory failure, renal failure resulting in dialysis, cardiac failure with cardiomyopathy and arrythmias, and enteric symptoms such as diarrhea and mesenteric ischemia.

Recent research identified mitochondrial dysfunction as an important factor in aging-related phenotypes. Mitochondrial physiological functions include oxidative phosphorylation, regulation of reactive oxygen species (ROS), ATP synthesis, thermoregulation, lipid and iron metabolism, autophagy, apoptosis, and regulation of hormones such as cortisol, estrogen, and thyroid hormones. In aged persons, the cells composing tissues have accumulated damage, mutations, and replication errors in mitochondrial DNA (mtDNA). Mitochondrial respiratory chains produce reactive oxygen species (ROS), such as hydrogen peroxide, superoxide and hydroxyl radicals, and they also eliminate these ROS with enzymes such as superoxide dismutase [17]. Tissues of aged persons have accumulated senescent cells with dysfunctional mitochondria with damaged mtDNA, which cannot keep up with excessive oxygen and metabolic demands associated with acute inflammation such as COVID-19 infection. Such hyperinflammatory conditions cause intracellular accumulation of ROS and ATP, and the failure to eliminate these toxic molecules enhances the damage to mitochondrial structures such as lipid membranes and biogenesis functions, further affecting glutathione and iron metabolism. The damage to the mitochondrial structure causes mtDNA release to systemic circulation, which works as damage-associated molecular patterns (DAMPs), resulting in further activation of innate immune reactions through toll-like receptors (TLRs) similar to pathogen-associated molecular patterns (PAMPs). Circulating mtDNA has been demonstrated as one of poor prognostic markers in ICU-related mortality as well as mtROS, heat shock protein (HSP), and high-mobility group homeobox protein 1 (HMGB1). DAMPs are also known to activate NF kappa B pathways and the stimulator of interferon genes (STING) pathway. Self DNAs and RNAs are the important DAMPs activating innate immune cells such as plasmacytoid dendritic cells and myeloid dendritic cells through TLR7, TLR9, and TLR3, respectively [17]. 

### 3.2. Host Response to SARV-COV-2 Determines the Prognosis of COVID-19

Lucas, Klein, and Iwasaki et al. [18,19] investigated patients with severe COVID-19 and compared the plasma cytokine levels and neutralizing antibody titers to SARS-CoV-2 between those who died and those who recovered from COVID-19. They found that elevation of plasma interferons (IFNs) in the early stage of the disease was higher in the patients who recovered from the disease, with elevated neutralization antibodies to the SARS-CoV-2 S protein, but an early increase in IFNs was not observed, and IFNs and neutralizing antibody titters increased in the late stage of disease in the patients who died from COVID-19. IFNs are among the important molecules preventing viral infection, and the authors considered that an increased expression of IFNs and neutralizing antibodies in the early phase of the infection may result in early and complete eradication of the SARS-CoV-2 virus, while a delayed increase in IFNs with a delayed elevation of neutralizing antibody titers is too late to eradicate the virus and further causes a late cytokine storm of the host, resulting in death. The difference in IFN response and neutralizing antibody titers in the early phase of the infection would be the essential factor in determining the prognosis of this disease, which may be genetically determined or affected by comorbidities, age, and/or concomitant therapeutics.

### 3.3. COVID-19 and Inflammatory Skin Disease

#### 3.3.1. Psoriasis

Psoriasis is one of the major inflammatory skin diseases which affects 2~3% of Western people and 0.2~0.3% of Japanese people. The pathogenesis of psoriasis has been increasingly elucidated due to the rapid development of biological treatments under the recent advancement in immunology, molecular biology, and biomedical techniques. Psoriasis is known for the increased expression of antimicrobial peptides, elevated Th17 and Th1 response, which may cause an increased defense against infectious attacks, including COVID-19. Psoriasis has also been known recently for its association with metabolic syndromes, such as obesity, diabetes mellitus (DM), hypertension, hyperglycemia, and cardiovascular diseases (CVD), as well as with other inflammatory conditions, such as inflammatory bowel disease (IBD), uveitis, arthritis, and psychomotor disorders [20]. These systemic conditions are related to upregulated inflammation and an increased risk of hospitalization caused by COVID-19 in the general population. However, the cohort of psoriasis patients suffering from COVID-19 did not show an increased risk of hospitalization as the above metabolic conditions in the population without skin diseases shows [21,22,23,24,25,26,27]. It could be because the number of patients has been small and the statistical significance is not obvious so far, but could be because of any unknown factors, such as genetic background, causing psoriasis patients to be less susceptible to severe infection. Another explanation could be that psoriasis patients are well-treated with anti-inflammatory agents, such as anti-TNF antibodies and other biologics and immunosuppressive agents, and that they are not at a higher risk of severe infection with COVID-19 [28]. 

Rheumatoid arthritis is another inflammatory condition often treated with biologics commonly used in psoriasis and psoriatic arthritis treatment. A Danish cohort study on immune-mediated inflammatory diseases (IMIDs) showed an increased risk of COVID-19-related hospitalization and mortality with rheumatologic disease, while those with dermatologic disease and gastrointestinal disease showed a decreased risk [27]. It may be due to the difference in genetic background of these diseases, treatment history, comorbidities, and concomitant treatments, but may be due to an unknown pathophysiological status of these diseases. Several independent studies showed that psoriasis patients under a biologics treatment did not show an increased risk of COVID-19 infection [27,28,29,30,31,32,33,34,35,36], or showed a decreased risk of severe COVID-19 infection [37], although an early study suggested an increased risk of mild-to-moderate COVID-19 infection in biologics-treated psoriasis patients [23]. TNF inhibitors plus MTX in psoriasis patients showed no difference in severe COVID-19 infection, but in certain cohorts [38,39], this treatment showed a decreased risk of COVID-19 infection and COVID-19-related mortality among IMID patients. Recent studies showed that non-biologics- or biologics-treated psoriasis patients with positive COVID-19 showed a decreased risk of severe COVID-19 infection [39,40,41]. These results suggest that IMIDs other than psoriasis treated with non-biologics may have an increased risk of severe COVID-19, while psoriasis patients are not at a higher risk of severe COVID-19 infection when treated either with biologics or non-biologics. In rheumatoid arthritis (RA), biologic DMARDs (bDMARDs), especially TNF inhibitors, decrease the risk of hospitalization, while rituximab and systemic corticosteroids increase the hospitalization risk; thus, the risk of severe COVID-19 infection also depends largely on treatment modalities.

Apremilast, one of the phosphodiesterase 4 inhibitors, increases the cellular concentration of cAMP by suppressing the phosphodiesterase activity, thus resulting in the suppression of inflammation. There have been some favorable reports on apremilast in the use for psoriasis patients in the COVID-19 era. Dommasch et al. reported that the intake of aprelimast decreased the risk of severe viral infection. The risk of infectious disease decreased in psoriasis patients treated with apremilast [42]. The risk of severe COVID-19 infection did not increase with the use of apremilast in psoriasis patients [43,44,45,46].

Glucocorticoids are sometimes used systemically in GPP patients and reduce the risk of severe COVID-19 infection, but on the other hand, they delay the clearance of SARS, MARS, and SARS-CoV-2 viruses from the blood or respiratory tract, and recent reports demonstrated an adverse effect of glucocorticoids in the COVID-19 infection [47].

Cyclosporin is often used in psoriasis patients in Japan, but there have been no accumulative data of psoriasis patients treated with cyclosporin with COVID-19 infection. Cyclosporin is used mostly in post-transplantation patients, and those post-transplantation patients under cyclosporin treatment did not negatively affect the course of the COVID-19 infection [48]. Psoriasis patients treated with non-biologics, including cyclosporin, did not show an increased risk of severe COVID-19 infection in several cohort studies [39,49,50,51]. The summary of the epidemiological studies previously reported is shown in Table 1.

The true mechanism is not yet clear, but the overall safety of biologics treatment in psoriasis patients seems to have been established in the era of COVID-19, and it is recommended that psoriasis patients on biologics treatment should continue their treatment during the COVID-19 outbreak. The recent guidelines published by several dermatological associations recommend that psoriasis patients should continue their treatment during the era of COVID-19 unless they are infected with SARS-CoV-2 to avoid exacerbation or flare of psoriasis, which would cause skin barrier disruption and make them susceptible to systemic infection such as sepsis through damaged skin [53,54,55]. 

#### 3.3.2. Atopic Dermatitis

Atopic dermatitis is another major chronic inflammatory skin disease, whose treatment has been changing according to the recent developments in immunology, molecular biology, and biomedical techniques as seen in psoriasis. Atopic dermatitis is considered a Th2-balanced disease often associated with bronchial asthma, allergic rhinitis, allergic conjunctivitis, and food allergy [55]. Atopic dermatitis has been reported to be associated with an increased risk of COVID-19 in some research [56,57,58] and with decreased risk [59,60,61] or no difference in other studies [60,61,62]. 

Dupilumab, the antibody against IL-4 receptor α, which blocks IL-13 and IL-4 signaling, brings back Th2-balanced immunity to the Th1-balanced direction and normalizes the immune status [63]. Th2 immunity usually tethers the immune defense against viral or bacterial infections, and the atopic dermatitis patients are vulnerable to infections such as herpes simplex and staphylococcus or streptococcus species, and bringing back to the Th1-balanced direction by dupilumab may theoretically strengthen the defensive action against infectious insults in atopic dermatitis patients [46]. Dupilumab has been shown to reduce the risk of serious infection, including skin infections and herpes simplex infection, by pooled analysis of double-blind placebo-controlled studies [64], and may thus result in reduced infection in COVID-19. Dupilumab has been shown in several cohort studies to reduce the risk of COVID-19 infection [61], although another report did not show a decreased risk [62]. A large retrospective study demonstrated a slightly increased risk of COVID-19 infection in adult AD patients, and dupilumab decreased it [55]. Dupilumab also reduced the risk of COVID-19 infection in asthma patients [62,65]. 

Asthma patients express an increased level of TMPRSS2, enzymes needed for SARS-CoV-2 entry to cells, in their nasal mucosa, and are thus considered vulnerable to its infections; on the other hand, the level of the ACE2 expression, the receptor for SARS-CoV-2, has been reported decreased according to the increased exposure to environmental allergens, which may protect them from SARS-CoV-2 infection. Asthma patients present similar symptoms as COVID-19, such as recalcitrant cough and dyspnea; thus, caution is needed to carefully distinguish the symptoms of exacerbation of asthma from COVID-19 [66]. In addition, virus infection, including COVID-19, can exacerbate asthma and other allergic disease symptoms, and telemedicine is recommended for those who do not need face-to-face medical examination during the COVID-19 pandemic [66]. 

Systemic corticosteroids are sometimes used in acute exacerbation in atopic dermatitis patients, but prolonged use of systemic corticosteroids has been reported to be associated with an increased risk of severe COVID-19 infection [67]. Acute exacerbation of atopic dermatitis makes patients susceptible to percutaneous infections. Zang et al. reported that moderate-to-severe AD patients are at a higher risk of COVID-19 infection and of having a more severe course. They are also at a higher risk of discontinuation of treatment and at higher anxiety levels [68]. As reported in psoriasis, atopic dermatitis patients are also recommended to continue their treatment, whether it is systemic or topical, to avoid acute exacerbation due to cessation of the treatment [69,70]. 

#### 3.3.3. JAK Inhibitors

JAK inhibitors are the emerging small-molecule agents to treat psoriasis and atopic dermatitis, in addition to rheumatoid arthritis, systemic lupus erythematosus, and IBD, and this class of drugs has also been approved for the treatment of cytokine storm in COVID-19 patients [71,72].

JAKs are cytoplasmic kinases which associate to cytokine receptors to transduce their signals to transcription factors, such as STATs, whose signaling pathway is called the JAK–STAT pathway (Figure 4). Cytokine receptors which can associate JAKs include IL-2, 3, 4, 5, 6, 7, 9, 10, 12, 13, 15, 22, 23, 31 and IFNs. JAKs are composed of four family members, JAK1, JAK2, JAK3, and Tyk2 [72]. Several specific inhibitors have been developed to block the specific JAKs to inhibit the specific signaling cascades in inflammatory diseases (Figure 5) [73]. Recent advancements in medicine have developed tofacitinib for rheumatoid arthritis and psoriasis (approved in the EU and the USA), baricitinib for RA and atopic dermatitis, upadacitinib for atopic dermatitis, ruxiolitinib for myelodysplastic syndrome, and several other JAK inhibitors are under clinical studies. This class of drugs can block IFN signals, which affects the defensive activity against viral infections, and may result in an increased risk of COVID-19 infection [18]. On the other hand, tofacitinib and baricitinib have been approved for the treatment of severe COVID-19 infection to decrease cytokine storm symptoms [74]. Baricitinib has been shown to have antiviral activity, inhibiting endocytosis and following viral replication in the cells, through blocking the effects of numb-associated kinases (NAKs) family members. Upadacitinib and tofacitinib do not have inhibitory effects on NAKs [75]. JAK inhibitors may be useful in treating the late stage, cytokine storm phase of COVD-19, but may increase the risk of infection of COVID-19; thus, caution is needed when treating patients with AD or psoriasis with this class of medications. However, no increased risk of COVID-19 infection, hospitalization, or mortality has been shown in the treatment with JAK inhibitors in atopic dermatitis, psoriasis, or rheumatic patients so far [76,77,78]. 

Nobari et al. [69] published a systematic review on COVID-19-infected skin disease patients treated with immunosuppressants. They reviewed 10 patients with psoriasis/psoriatic arthritis treated with biologics or immunosuppressive drugs including cyclosporin and methotrexate, and 2 patients with immuno-bullous disorders, including 1 pemphigus and 1 mucous pemphigoid patient. They reported that one patient did not stop biologics during the COVID-19 infection, and others did, but the clinical course of psoriasis was preferable, while 1 patient treated with cyclosporin and methotrexate experienced severe exacerbation after discontinuation of the previous treatment. Out of these 12 patients, only one patient needed hospitalization. They concluded that dermatologic disease patient treated with biologics had favorable clinical course after the discontinuation of the treatment during COVID-19 infection. Another report by Patrick et al. [79] demonstrated their epidemiological survey with gene expression analysis that skin diseases are associated with increased risk of COVID-19 infection, but related to less severe disease course. They found similar gene expression in skin diseases and COVID-19 infection, and speculated that inflammatory skin conditions such as atopic dermatitis and psoriasis have heightened immune response even in normal looking skin, and also in oral mucosa and throat, which may allow increased risk of infection but suppress viral activity with already heightened immune reaction.

The other possibility we propose is that risk for severe COVID-19 is highly organ-specific, and that skin inflammation is not the risk for severe COVID-19, but lung, heart, and vascular inflammation are the risk factors. Patients with skin inflammatory disease may have milder inflammation in lung, heart and vascular system compared to those without skin inflammatory diseases. 

The mechanism has not been elucidated, but epidemiological study so far demonstrates that dermatologic diseases have lower risk of severe COVID-19 infection and favorable course when infected with COVID-19. Especially patients with Rheumatic diseases have higher risk of severe COVID-19 infection compared to those with psoriasis and atopic dermatitis, treated with biologics, from the investigations performed so far [52,80]. However, atopic dermatitis has been reported to have increased risk of COVID-19 infection in some reports, and the outcome of COVID-19 infection may differ among different skin conditions, which needs future studies to establish precise risk of COVID-19 infection and severity in each dermatologic condition. 

## 4. Conclusions

COVID-19 is the most recent emerging infectious disease, which has plunged the whole world into chaos. The rapid and effective control of this infectious disease is the hope of all over the world. How and where SARS-CoV-2 emerged is another concern. If rapid development and deforestation of rich nature in outback continent of China resulting in host conversion of the virus, was one cause of the emergence of SARS-CoV-2, novel similar emerging infectious diseases will appear in near future. The globalized world will easily spread the emerging virus anywhere in the world. We should be aware of that and be prepared to the risk of further new emerging infectious disease and its pandemic.

The reason why dermatologic inflammatory diseases have better prognosis compared to systemic inflammatory diseases, such as obesity, diabetes, hypertension and aging is elusive. Some possible reasons can be postulated, but they are merely speculations. There is no conclusive data on the discrepancy between cutaneous inflammatory diseases and systemic inflammatory diseases, the speculation could be that cutaneous inflammatory diseases, even if they have systemic inflammation, the main site of inflammation is located on skin, which may have different impact on COVID-19 infection derived from organ specificity. Further investigation is needed to elucidate this difference.

## Figures and Tables

**Figure 1 ijms-23-12260-f001:**
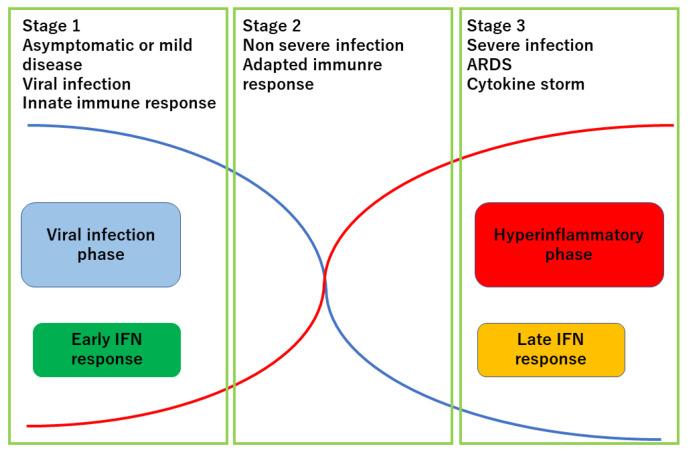
Three stages of COVID-19 infection. Modified from reference [8].

**Figure 2 ijms-23-12260-f002:**
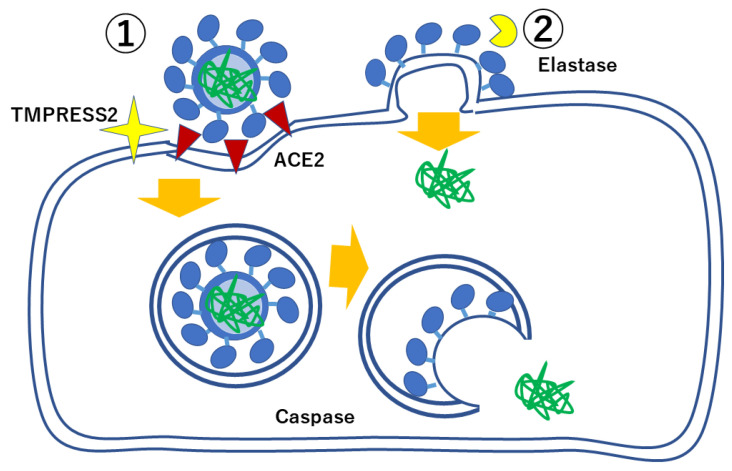
The infection pathway of SARS-CoV-2 differs depending on the inflammation status. Without inflammation, SARS-CoV-2 utilizes TMPRSS2 to enter cells with the ACE2 receptor through endocytosis, while with inflammation, SARS-CoV-2 utilizes environmental elastases to enter cells through membrane fusion, with rapid replication of themselves. Modified from reference [13].

**Figure 3 ijms-23-12260-f003:**
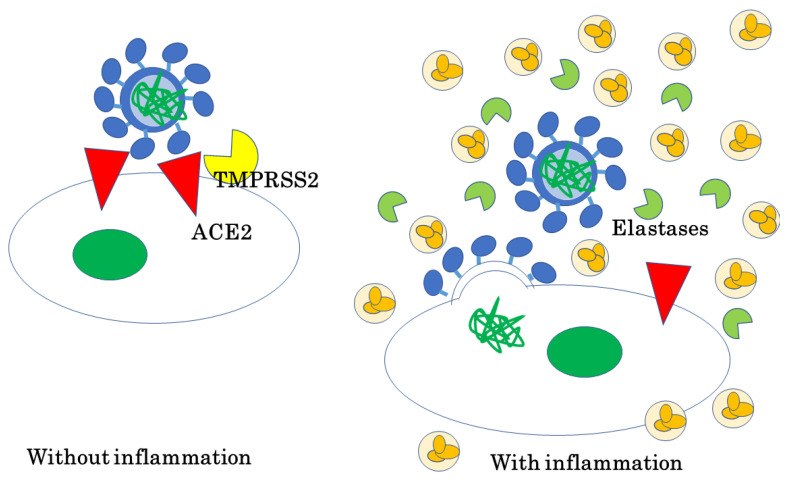
SARS-CoV-2 and SARS-CoV utilize ACE2 as their receptor with no inflammation, while they utilize abundant environmental elastases with inflammatory condition to enter cells.

**Figure 4 ijms-23-12260-f004:**
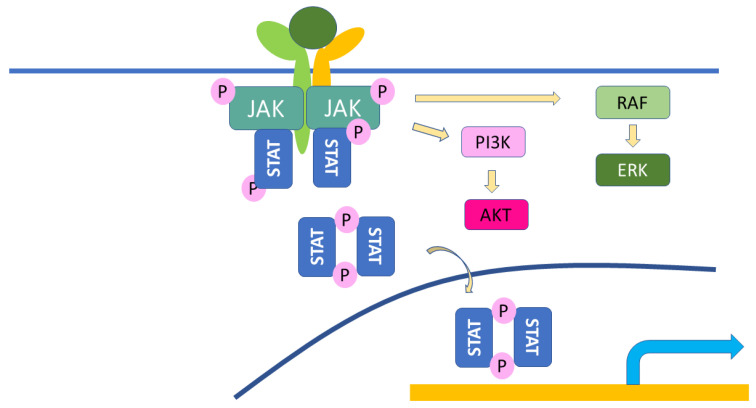
The scheme of the JAK–STAT pathway. When a cytokine binds to its receptor, the receptor makes a receptor complex, which phosphorylates JAK, and JAK phosphorylates itself and STATs. Phosphorylated STATs make homo- or heterodimers and translocate in the nucleus, where they exert their function as nuclear proteins. Modified from reference [71].

**Figure 5 ijms-23-12260-f005:**
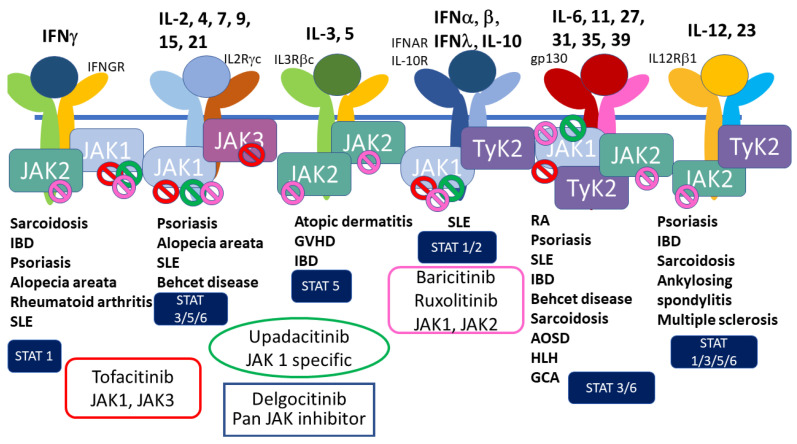
Several cytokine receptors utilize the JAK–STAT pathway to transduce their signals to the nucleus. Different cytokine receptors utilize different combinations of JAK family members and phosphorylate different STAT family members. Modified from reference [71].

**Table 1 ijms-23-12260-t001:** Published epidemiological studies on COVID-19 and psoriasis.

Authors	Journal, Year of Publication	Number of Patients Investigated	Conclusion
Penso et al. [40]	Br J Dermatol, 2022	1,326,312 psoriasis patients (France)	Systemic treatment including non-biologics and biologics of psoriasis patients did not increase the risk of in-hospital mortality due to COVID-19
Gisondi et al. [30]	J Allergy Clin Immunol, 2021	6501 psoriasis patients (Italy)	Biologics treatment did not increase the risk of hospitalization or mortality in psoriasis patients
Talamonti et al. [52]	Expert Opin Biol Ther, 2021	12,807 psoriasis patients (Italy)	Psoraisis showed a comparable infection rate compared to the general population
Mahil et al. [37]	J Allergy Clin Immunol, 2021	374 psoriasis infected with COVID-19 (International)	Patients treated with biologics showed a decreased risk of severe COVID-19 infection with a lower hospitalization rate
Baniandres-Rodoriguez et al. [31]	J Am Acad Dermatol, 2021	2329 psoriasis patients on systemic treatment (Spain)	Comparable risk of hospitalization and mortality compared to the general population
Yousaf et al. [38]	J Am Acad Dermatol, 2021	53 million people (worldwide)	No evidence of increased risk of hospitalization in the patients prescribed TNFi and/or MTX
Belleudi [26]	J Clin Med, 2021	22,406 psoriasis patients	No increased risk of hospitalization or death related to COVID-19
Eder et al. [25]	Arthritis Care Res (Hoboken), 2021	1505 psoriasis patients, 111 PsA patients (OHIP database, US)	Comparable risk of hospitalization
Attauabi et al. [27]	J Autoimmun, 2021	5305 AD, 8784 PSO, 4160 PsA (Denmark)	Lower risk of COVID-19 infection in AD, comparable risk in PSO and PsA
Gisondi et al. [51]	Vaccines, 2020	Systematic literature review on 27 references	Psoriasis patients treated with systemic treatment did not show an increased risk of COVID-19 infection. Other IMIDs including RA and IBD treated with biologics show similar clinical outcoms of COVID-19
Piaserico et al. [36]	Am J Clin Dermatol, 2020	1830 psoriasis patients	No increased risk of severe COVID-19 infection in psoriasis patients on biologics
Ciechanowicz et al. [32]	J Dermatol Treat, 2022	61 patients	Biologics treatment was not associated with an increased risk of severe COVID-19 infection in psoriasis patients
Fougerousse et al. [44]	J Eur Acad Dermatol Venereol, 2020	1418 psoriasis patients	No increaed risk of severe COVID-19 in psoriasis patients treated with systemic or biologics treatment
Damiani et al. [23]	Dermatol Ther, 2020	1193 PSO patients, 10,060,574 controls	Increaed risk of mild-to-moderate COVID-19 infection in psoriasis patients treated with biologics
Fulgencio-Barbarin et al. [50]	Int J Dermatol, 2020	465 psoriasis patients in Madrid	No increased risk of severe COVID-19 in psoriasis and systemic treatment
Ekinci et al. [33]	Dermatol Ther, 2020	133 psoriasis patients with biologics treatment in Turkiye	No increased risk of COVID-19 infection and its severity in biologics-treated psoriasis patients
Brazelli et al. [34]	Dermatol Ther, 2020	180 psoriasis patients with topical treatment alone (100) and biologics treatment (80)	No increaed risk of moderate-to-severe COVID-19 infection in biologics-treated psoriasis patients
Simon et al. [41]	Nat Commun, 2020	534 immunodulatory inflammatory disease patients with cytokine inhibitors, 259 IMID patients treated with non-cytokine inhibitors, 285 healthcare controls, 971 non-healthcare controls	IMID patients treated with cytokine inhibitors showed a lower prevalence of positive COVID-19 serum antibodies
Camela et al. [35]	Dermatoly, 2020	Interviews with 965 psoriasis patients in Naples	No increased risk of severe COVID-19 in psoriasis patients treated with biologics
Gisondi et al. [24]	J Am Acad Dermatol, 2020	980 psoriasis patients compared to 257,353 people (Verona population)	No increased hospitalization or death in psoriasis patients

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
