# Peer review of "Inflammation Causes Exacerbation of COVID-19: How about Skin Inflammation?"

_ijms, 2022, doi:10.3390/ijms232012260_

Round 1

Reviewer 1 Report

A minor spell check is required for English language 

Accepted for publication

Author Response

To reviewer 1

Thank you very much for your kindly reviewing our manuscript.

We revised our manuscript following your comments.

A minor spell check is required for English language 

Response: We checked spelling and corrected mis-spelling in the text.

Reviewer 2 Report

Discuss Some points

1.  How does SARS-CoV-2 induce excessive inflammation through the activated RAS signaling pathway?

2. A recent study identified that the three target proteins (MAPK8, MAPK10, and BAD) are mainly associated with the RAS signaling pathway and they are the key mediators of inflammation, vasoconstriction, and thrombosis. doi.org/10.1038/s41598-021-88313-5

Discuss how these target proteins are connected and contribute to the deterioration of the inflammation condition.

Author Response

To reviewer 2

Thank you very much for your time to review our manuscript.

We have revised our manuscript following your comments.

  1. How does SARS-CoV-2 induce excessive inflammation through the activated RAS signaling pathway?
  2. A recent study identified that the three target proteins (MAPK8, MAPK10, and BAD) are mainly associated with the RAS signaling pathway and they are the key mediators of inflammation, vasoconstriction, and thrombosis. doi.org/10.1038/s41598-021-88313-5

Discuss how these target proteins are connected and contribute to the deterioration of the inflammation condition.

Response: Thank you very much for your important suggestions. As you suggested, COVID19 infection activate RAS signaling by internalizing ACE2, and through MAPK8, MAPK10 and BAD, probably by inducing autophagy. We have added the following sentences in the text.

Recent findings reveled that Renin-Angiotensin Systems (RAS) pathway, the main endocrine pathway causing hypertension, is involved in COVID19 inflammation. The internalization of the virus downregulates the ACE2 expression on the cell surface, which causes elevation of angiotensin II. Subsequent overactivation of cardiovascular system, induces vasoconstriction, increases blood pressure and thrombosis. Recent studies revealed that hypertensive heart injury is dependent on dysregulation of autophagy, which is considered the novel therapeutic target of cardiovascular diseases. Angiotensin II has been shown to induce increase in autophagy, and in vivo treatment with an autophagy inhibitor decreased blood pressure and improved endothelium-dependent relaxation. MAPK8 (JNK1) is involved in autophagy by phosphorylation of BCL2 (B cell leukemia/lymphoma 2), disrupting the interaction of BCL2 and BECN1 (beclin 1), resulting in BECN1-dependent autophagy. Recent network pharmacology approach revealed the novel therapeutic targets of COVID19 inflammation, including MAPK8 (JNK1), MAPK10 (JNK3) and BAD (Bcl-2-associated death promoter), which are involved in RAS signaling pathway and autophagy. The inhibitors of these molecules, by decreasing RAS activation and autophagy, downregulate cardiovascular burden, vasoconstriction and thrombosis.   
